# Exploring the effects of degraded vision on sensorimotor performance

William E. A. Sheppard[1]*, Polly Dickerson[2], Rigmor C. Baraas[3], Mark Mon-Williams[1,3,4], Brendan T. Barrett[5], Richard M. Wilkie[1], Rachel O. Coats[1]

1 School of Psychology, University of Leeds, Leeds, West Yorkshire, United Kingdom, 2 Department of Ophthalmology, York Teaching Hospital NHS Foundation Trust, North Yorkshire, United Kingdom, 3 Department of Optometry, Radiography and Lighting Design, National Centre for Optics, Vision and Eye Care, University of South-Eastern Norway, Kongsberg, Norway, 4 Bradford Institute of Health Research, Bradford Teaching Hospital NHS Foundation Trust, West Yorkshire, United Kingdom, 5 Faculty of Life Sciences, School of Optometry & Vision Science, University of Bradford, West Yorkshire, United Kingdom

* w.e.sheppard@leeds.ac.uk

## Abstract

### Purpose

Many people experience unilateral degraded vision, usually owing to a developmental or age-related disorder. There are unresolved questions regarding the extent to which such unilateral visual deficits impact on sensorimotor performance; an important issue as sensorimotor limitations can constrain quality of life by restricting 'activities of daily living'. Examination of the relationship between visual deficit and sensorimotor performance is essential for determining the functional implications of ophthalmic conditions. This study attempts to explore the effect of unilaterally degraded vision on sensorimotor performance.

### Methods

In Experiment 1 we simulated visual deficits in 30 participants using unilateral and bilateral Bangerter filters to explore whether motor performance was affected in water pouring, peg placing, and aiming tasks. Experiment 2 (n = 74) tested the hypothesis that kinematic measures are associated with visuomotor deficits by measuring the impact of small visual sensitivity decrements created by monocular viewing on sensorimotor interactions with targets presented on a planar surface in aiming, tracking and steering tasks.

### Results

In Experiment 1, the filters caused decreased task performance—confirming that unilateral (and bilateral) visual loss has functional implications. In Experiment 2, kinematic measures were affected by monocular viewing in two of three tasks requiring rapid online visual feedback (aiming and steering).

### Conclusions

Unilateral visual loss has a measurable impact on sensorimotor performance. The benefits of binocular vision may be particularly important for some groups (e.g. older adults) where

**Data Availability Statement:** All 2 files are available from the GitHub database (https://github.com/willsheppard9895/DegradedVisionData.git).

**Funding:** This study was funded by the Medical Research Council's ActEarly program through a grant awarded to MM (MR/S037527/1).

**Competing interests:** The authors have declared that no competing interests exist.

an inability to complete sensorimotor tasks may necessitate assisted living. There is an urgent need to develop rigorous kinematic approaches to the quantification of the functional impact of unilaterally degraded vision and of the benefits associated with treatments for unilateral ophthalmic conditions to enable informed decisions around treatment.

## Introduction

It has been estimated that 11% of older adults in the US have an interocular difference in vision of three or more lines [1]. Throughout the manuscript, we use the term vision to refer to both unaided vision (no refractive correction), and aided visual acuity (with refractive correction). These differences in vision may persist despite optimal refractive correction and can be attributed to a range of factors, including developmental disorders (e.g. amblyopia) and age-related change (e.g. cataract, macular degeneration). These ophthalmic conditions can often be treated, but all forms of treatment have costs (in terms of funding or patient time) and there are often risks associated (e.g. surgical complications). To make informed decisions regarding the treatment of these conditions, we must first understand the potential benefits to the patient such as improvements to their health and lifestyle. The extant literature often focuses on *self-reported* measures of quality of life and/or functional abilities [2–5]. Self-report measures have brought a lot of value to research, and serve a particular purpose, however, these are open to bias and often struggle to accurately quantify the impact of degraded vision upon specific motor skills. This raises the question: what is the functional impact of a unilateral visual deficit?

The present study set out to test issues relating to the impact of degrading vision in one eye (unilaterally degraded vision), and the related impact on binocular vision and, thus, functional motor behaviours. It stands to reason that unilaterally degraded vision may impact motor behaviours as binocular viewing confers many advantages over monocular viewing. For example, extraretinal vergence information can help specify the distance of objects and is used during the planning of reaching movements [6,7]. Two eyes afford a wider field of view compared to one eye alone and are associated with improved perceptual sensitivity through binocular summation [8–11]. Binocular viewing also facilitates stereopsis–the gathering of perceptual information available through horizontal retinal disparities. There are strong grounds for supposing that stereopsis confers an important functional advantage. Stereopsis has been shown to contribute to the perception of shape geometry, including depth [12,13], slant [14], and curvature [12,15,16], information that is important when grasping objects.

The theoretical advantages of binocular vision and the empirical demonstration of improved perceptual sensitivity during binocular viewing suggest that reductions in binocular vision will have functional significance. But there is a scientific need to test this conjecture and establish empirically the impact of reduced vision on the sensorimotor abilities that underpin activities of daily living (ADLs). Although our ultimate goal is to examine the effects of visual deficits on sensorimotor performance, particularly in older adults for whom visual deficits are more common, we first took a more conservative approach of testing young, healthy adults. If we can detect and measure functional impairment in healthy individuals with good vision (by allowing only monocular viewing or by degrading vision in one eye during binocular viewing) then this justifies running large scale studies to examine the impact of unilateral ophthalmic deficits in patient groups.

Many studies have explored whether monocular viewing conditions have functional significance. For example, catching performance has been shown to be better with two eyes

compared to one, predominantly due to stereopsis providing spatial and temporal advantages [17–19]. Prehension is also impaired under monocular viewing conditions, with reaching and grasping performance significantly worse when only one eye is used [20–23].

Motor performance can also be impaired when monocular vision is degraded, rather than occluded. This was demonstrated by Piano and O'Connor, who systematically altered vision using convex spherical lenses whilst participants performed a water-pouring task, as well as a small and large bead threading task. Compared to baseline, any decrease in monocular vision impaired performance on both bead threading tasks, however, changes in water-pouring performance were only seen when comparing unimpaired vision to complete monocular suppression [24]. Vision can also be degraded using Bangerter filters, which decrease vision without changing the mean luminance of a stimulus [25]. Bangerter filters (0.2 logMAR neutral filters) have been used to degrade vision in one or both eyes of experienced basketball players. Monocular degradation was found to significantly reduce the number of successful free-throws when compared to full binocular vision [26]. Bangerter filters have also been shown to significantly impair the perception of features of walkways such as ramps and steps [27]. Difficulties in this area are associated with increased fall risks in older adults [28].

Reduced sensorimotor performance is also observed in those with naturally occurring monocular visual impairment due to ophthalmic conditions (e.g. amblyopia and cataract). Individuals with normal binocular vision reliably show better performance on motor tasks when compared to individuals with amblyopia—with the greatest differences in time-limited and/or novel tasks [29–31]. In a study comparing performance in a sixteen-item 'fine-motor skill' battery, children with amblyopia performed worse than controls on nine of the tasks, with the largest differences observed in tasks requiring speed and accuracy (e.g. drawing straight lines and peg placement). This result was not due to a subset of amblyopes with particularly poor performance but was driven by a general reduction in performance [32]. Sensorimotor performance is also hampered by unilateral cataract. Longitudinal studies in Vietnam and Australia have found decreases in falls and motor crashes following First-Eye cataract removal Surgery (FES), and further reductions following Second-Eye cataract removal Surgery (SES)–suggesting there are benefits of full binocular vision and SES [33,34].

In summary, there is strong convergent evidence to suggest that unilateral degraded vision has an impact on sensorimotor function in laboratory tasks, which in turn suggests a negative impact on an individual's ability to undertake ADLs, and therefore on their quality of life. These observations provide a rationale for the treatment of unilateral ophthalmic disorders even when the condition does not prevent the successful completion of visual tasks *per se* (such as driving). Unfortunately, there is a lack of studies that have accurately quantified the association between unilateral visual deficit and functional performance on sensorimotor tasks (despite the large number of studies that have established a relationship) [2–5,24,29,30,32,35].

In Experiment 1, we sought to explore the functional benefits of binocular vision. To do this, we replicated the typical approach found in the experimental literature by selecting sensorimotor tasks that relate to ADLs and then simulating visual loss. Visual loss was simulated using unilateral and bilateral Bangerter filters to create three visual conditions: no impairment (no filters), unilateral impairment and bilateral impairment. It was predicted that all scores on the visual measures would be best in the 'no impairment' condition. However, the hypothesis-driven prediction was that visual acuity (VA) and contrast sensitivity (CS) would be worst in the bilateral impairment condition and best with no impairment, but that stereopsis would be worst in the unilateral impairment condition (given the large interocular difference created by one filter). Unilateral and bilateral impairment were investigated as these have been shown to have differing effects on sensorimotor performance [26,33,34]. We sought to replicate the water-pouring task as per Piano and O'Connor [24], however, we extended our analysis to

include a composite measure of time and accuracy as unilateral visual deficits are associated with decreases in the speed and accuracy of movements [29–31]. We also explored an aiming task using the same planar task as Domkin et al, performance on which has previously been shown to correlate with binocular VA [36]. The Purdue Pegboard Task was also used to assess motor performance, as it would be in a clinical setting.

In terms of how the filters would impact performance on the sensorimotor tasks, the body of evidence relating unilaterally degraded vision to sensorimotor performance led to the expectation that there would be an impact of viewing condition across these tasks—with performance being worst in the bilateral impairment condition and best in the no impairment condition. There were two likely outcomes with the unilateral impairment condition–it would either cause performance to fall between the levels associated with bilateral impairment and no impairment, or result in no notable performance difference from the no impairment results—but it was not possible to predict *a priori* what pattern of results would be found.

Experiment 2 tested the hypothesis that the degradation of vision caused by monocular viewing would cause reduced performance on a task carried out on a planar surface that required rapid online visual feedback corrections. It can be seen that Experiment 2 tested the corollary hypothesis that kinematic measures can detect functional changes in sensorimotor performance of the magnitude created by visual degradation.

## Experiment 1

### Methods

**Participants.** Thirty undergraduate students (27 females) from the University of Leeds participated in this study and were recruited by opportunity sampling through the University of Leeds participant pool. Participants were 18 years old or older (*range* = 18–23, *mean* = 19.37, *SD* = 1.22), five participants were left-handed and they all self-reported having normal or corrected-to-normal (with contact lenses) vision. Participants were excluded if their vision was corrected using glasses as these would get in the way of the filters they were required to wear during the experiment. No participants reported any history of tremors or impaired motor or neurological function that might affect their ability to perform the tasks. Two participants were removed from the aiming task data analysis (leaving a total of 28 participants) as their performance was especially poor (their results were over two *SDs* from the *mean*) on the first condition they completed indicating they had not followed task instructions. Participants received four course credits as compensation for their participation. Ethical approval was granted by the University of Leeds Research Ethics Committee (Ethics Reference Number: PSC-192).

**Materials.** All participants wore 0.3 logMAR Bangerter filters to impair VA to around the minimum level required for driving in the UK [37] (although the resulting level of acuity would be affected by their normal vision level). Filters were attached to one or both lenses of non-prescription glasses. We examined VA, CS and stereopsis to determine the effect of the filters on the participant's vision. In the unilateral impairment condition, the filter was placed over the participant's non-dominant eye. To establish eye dominance, participants completed the 'alignment test' [38]. Participants were asked to align their dominant index finger with a dot on the wall (approximately 6 feet from the participant). The subject was then asked to close their right eye, if the finger and the dot are still reported as aligned, the participant is left eye dominant. This procedure is then repeated for the left eye, if the finger and the dot are still reported as aligned, the participant is right eye dominant [38]. All visual measures and sensorimotor tasks were completed with both eyes open, and either no Bangerter filter applied to the glasses (no impairment), a filter applied to the non-dominant eye (unilateral impairment) or a filter applied to both eyes (bilateral impairment).

VA was measured using a logMAR chart [39]. This chart consisted of 14 rows each with five letters equally distanced from each other and of identical legibility [40]. Participants stood 6m from the chart and were instructed to read the letters from each row until the letters became too small for them to read. A per-letter scoring system (0.02 log units per letter) was used, with a lower score demonstrating better VA.

CS was assessed using the standardized Hamilton-Veale Contrast Sensitivity Test [41]. Participants stood 1m away from a wall-mounted chart (located at eye-height) consisting of 16 identically spaced and sized pairs of letters over 8 rows. The letters varied in contrast only, with an increment of 0.15 log units, ranging from 0 to 2.25 log units. Participants were instructed to read aloud as many letters as they could, and the higher the score, the better the CS [42].

Stereopsis was assessed using the Wirt Circles test from the standardized Titmus Stereo Fly Test [43]. Participants were presented with a chart containing nine diamond shapes; there are three rows of three diamonds, with each diamond containing four circles, one in each corner. The Wirt circle targets allow a near-threshold measurement of stereoacuity, with each set providing a different level of disparity down to 40 seconds of arc [43]. The chart was held 40cm from the participants' eyes and the participant was asked to state which of the four circles appeared to be the closest. This was performed while wearing a pair of polarised glasses over the non-prescription glasses. A disparity is created from the different information presented to the right and left eyes. Participants who correctly identified 1 circle were deemed to have a maximum stereoacuity of 800 seconds of arc, whereas those who correctly identified all 9 circles were deemed to have a maximum stereoacuity of 40 seconds of arc.

For water-pouring, as used by O'Connor et al [30], participants were seated 30cm from five horizontally aligned plastic beakers. Participants were required to fill each beaker to a 40ml mark as quickly and accurately as possible using a jug containing 500ml of water. Participants were not allowed to touch the beakers or to refill a beaker on a second attempt. Absolute accuracy (the total volume of water either above or below the 40ml marker) and time in seconds (from the time the jug was lifted to when it was replaced) were recorded. Time was measured using a stopwatch and was recorded to the closest hundredth of a second. Lower scores in both measures were indicative of more accurate and faster water-pouring, respectively. Many real-world tasks are both time-bound and require high levels of precision e.g. wood/metal work or using a mouse to operate a computer, therefore, to capture the effect of degraded vision on such tasks a composite measure of water-pouring performance was calculated using multiplication of absolute accuracy and time, and again, a lower composite score indicated an overall better water-pouring performance.

The Purdue Pegboard is a standardized measure of motor skill [44,45]. Sitting directly in front of the Purdue Pegboard, participants were instructed to use their preferred hand, taking one peg at a time and placing them into the designated row of holes. Participants were required to place as many pegs as possible in a period of 30 seconds, starting with the hole furthest away from them. Participants were not permitted to replace dropped pins. The total number of correctly placed pins was recorded, with a high number demonstrating better motor skills.

Participants also completed an aiming task presented within the Clinical Kinematic Assessment Tool, CKAT (Fig 1): a tool that provides objective measures of sensorimotor performance [46]. CKAT presents interactive visual stimuli on a tablet laptop screen whilst recording participants' kinematic responses to these stimuli. The CKAT was presented on Toshiba portable tablet computers (Portege M700-13P, screen size = 303x190mm, resolution = 1280x800, estimated visual angle = 40.56x26.10 degrees). Visual stimuli were refreshed at 60Hz and movement data was sampled at 120Hz and a 10Hz dual-pass Butterworth filter was applied to movement data at the end of each session. A pen-shaped stylus (140 x 9 mm) was

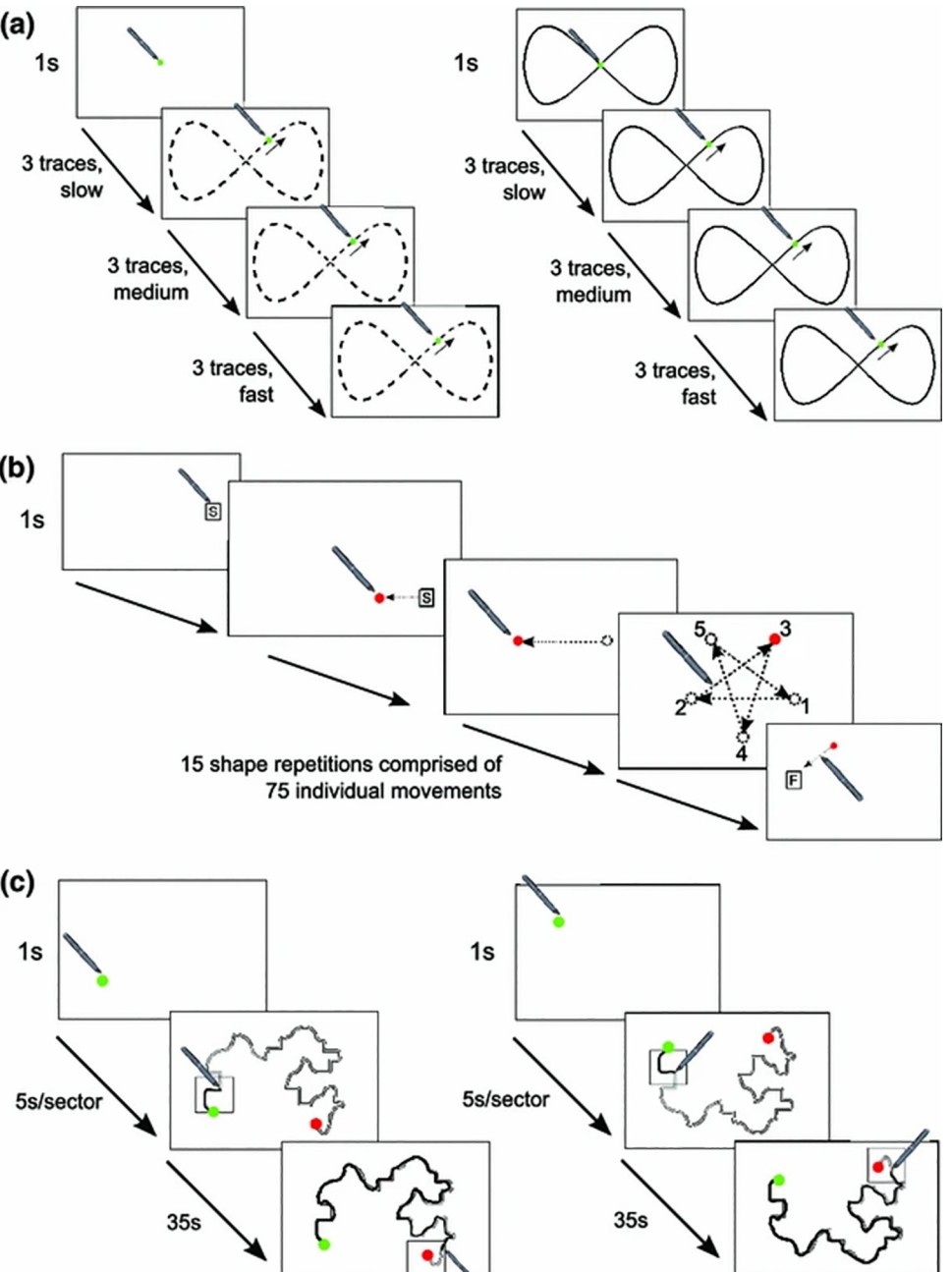

**Fig 1.** Illustration of the three motor control tasks: a) tracking, b) aiming and c) steering. a) Left panel shows the No Guide (NG) tracking condition, the dotted line indicates the trajectory of the moving dot. The right panel demonstrates the With Guide (WG) tracking condition, the solid line is the spatial guideline. b) Shows the aiming task. The red dot is the target, the arrows signal the movements that participants would make with the stylus between target locations and the numbers indicate the sequence in which the targets appeared. c) The left panel shows the primary path steering condition and the right panel shows the alternative path (mirror image) condition. The light grey line is the 'ink trail' showing the path taken. Participants follow the shape with their stylus and they are instructed to stay within the box shown, the box moves every 5 seconds. (Image adapted with permission from Flatters et al [47]). Note, only the aiming task (panel b) is completed during Experiment 1, whereas all three tasks (panel a-c) are completed in Experiment 2 and, therefore, are included here for future reference.

used as the input device. The task was presented on the tablet screen in landscape orientation on a level table in front of the participant. The CKAT software records stylus position to capture various kinematic measures (e.g. movement time, which we present here) to provide information about the accuracy and efficiency of participants' movements (see Culmer et al. [46] and Flatters et al. [47] for a full overview of the CKAT software and tests). In Experiment 1, participants only completed the aiming task (Fig 1, panel b). For the aiming task, participants were instructed to move as quickly and accurately from one target (5 x 5mm diameter dot presented sequentially 113mm apart) to another. Participants were required to keep the stylus in constant contact with the screen while moving it as quickly as possible from the start position to a green dot on the screen. Upon reaching the dot it disappeared and a new dot appeared elsewhere on the screen, which the participant then moved to. The physical separation of successive dots (targets) on the screen was the same. The participant made a total of 75 movements taking approximately 2–4 minutes. The time from leaving one dot (or the start box) and arriving at the next dot (or the finish box; movement time [MT]) was recorded. The median MT for the first 50 movements is reported here, with lower scores indicative of better performance. Only the first 50 were analysed as the final 25 trials contain several trials in which the dot jumps after a movement is initiated; which produces behaviour that is not relevant for the current study.

All measures were taken with both eyes open. Participants completed each task once with no opportunity for practice, however, tasks were straightforward and easily learned. As stated below, the order of the visual condition was counterbalanced between participants to mitigate for practice effects, so some participants started with unilateral impairment, some with no impairment etc. All participants also had the opportunity to ask questions if instructions were not clear.

**Procedure.** After consenting, participants provided demographic, handedness and vision (corrected where contact lenses are worn, or uncorrected if no glasses or contact lenses are worn) information. All participants completed the tasks in the same order, but the order of the visual condition was counterbalanced between participants (Fig 2).

**Design and data analysis.** A repeated-measures design was used with the visual condition (no impairment, unilateral impairment, bilateral impairment) as the independent variable for all measures. The statistical software package JASP [48] was used to conduct repeated-measures ANOVAs to examine differences between each level of visual condition for each visual and motor measure. Where Mauchly's test of sphericity was violated, a Greenhouse-Geisser correction was applied. Where it was not possible to compute an ANOVA statistic, due to non-parametric data structure, a Friedman One-Way Repeated Measure Analysis of Variance by Ranks was carried out. Planned pairwise comparisons were used to explore significant main effects of hypothesised condition differences, and a Bonferroni Holm correction was applied. Effect sizes (*Cohen's d*) for non-parametric analyses were calculated *as per* Rosenthal [49,50]. Non-parametric analysis was performed used RStudio Version 1.3.959 [51]. All data are available from the GitHub database (https://github.com/willsheppard9895/DegradedVisionData.git).

## Results

**Visual measures.** *Visual acuity.* Mean VA (logMAR) was worst in the bilateral impairment condition and best in the no impairment condition (Fig 3A). A significant main effect of visual condition emerged ($F_{2,58} = 293.37$, $p < .001$, $\eta^2 = .91$). This was driven by significant differences in logMAR scores between the no impairment and unilateral impairment conditions ($p = .037$, *Cohen's d* = .40), no impairment and bilateral impairments ($p < .001$, $d = 3.63$) and between unilateral impairment and bilateral impairment ($p < .001$, $d = 3.51$).

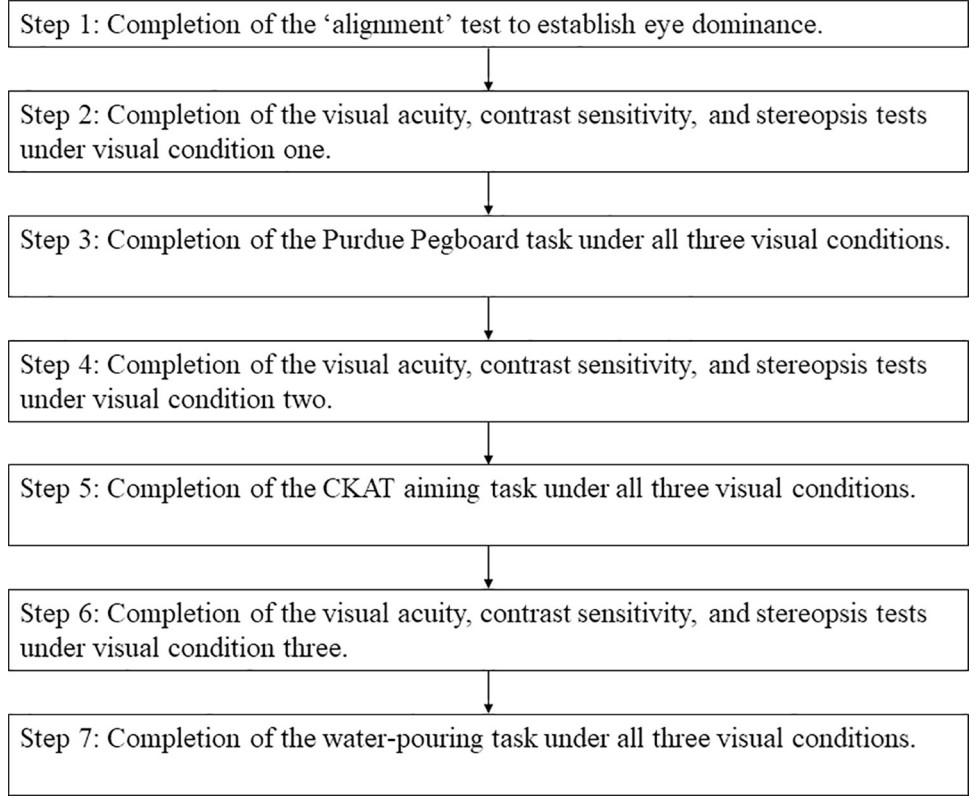

**Fig 2. Experimental procedure flowchart.** The order of visual condition completed (unilateral, bilateral and no impairment) was counter-balanced, so condition one refers to no impairment for some participants, unilateral impairment for others and so on.

*Contrast sensitivity*. Mean CS scores (see Fig 3B) were worst in the bilateral impairment condition and best in the no impairment condition. A significant main effect of visual condition was found ($F_{r2}$ = 47.59, $p < .001$, *Kendall's W* = .79). This was driven by significant differences in scores between no impairment and unilateral impairment ($p < .001$, $d = .1.42$), no impairment and bilateral impairment ($p < .001$, $d = 2.62$), and between unilateral impairment and bilateral impairment ($p < .001$, $d = 2.58$).

*Stereopsis*. One participant was removed as their data was incorrectly coded at the point of collection, leaving a participant total of 29. Mean stereopsis scores (see Fig 3C) were worst in the unilateral impairment condition and best in the no impairment condition. A significant main effect of visual condition emerged ($F_{r2}$ = 41.13, $p < .001$, *Kendall's W* = .71). This was driven by significant differences in scores between no impairment and unilateral impairment ($p < .001$, $d = 2.16$), no impairment and bilateral impairment ($p < .001$, $d = 2.16$), and between unilateral impairment and bilateral impairment ($p = .006$, $d = 1.05$).

**Motor measures.** *Water-pouring*. Water-pouring time scores are shown in Fig 4A. There was no main effect of visual condition on water-pouring time ($F_{2,58}$ = 1.95, $p = .152$, $\eta^2 = .06$). For performance in terms of accuracy, participants were most accurate in the no impairment condition and least accurate in the bilateral impairment condition (Fig 4B) and a significant main effect of visual condition was found ($F_{2,58}$ = 5.79, $p = .005$, $\eta^2 = .17$). This was driven by a significant difference in accuracy between no impairment and bilateral impairment ($p = .010$, $d = .586$). The composite measure of speed*accuracy showed the best scores under the no impairment condition and the worst scores under the bilateral impairment condition (Fig 4C).

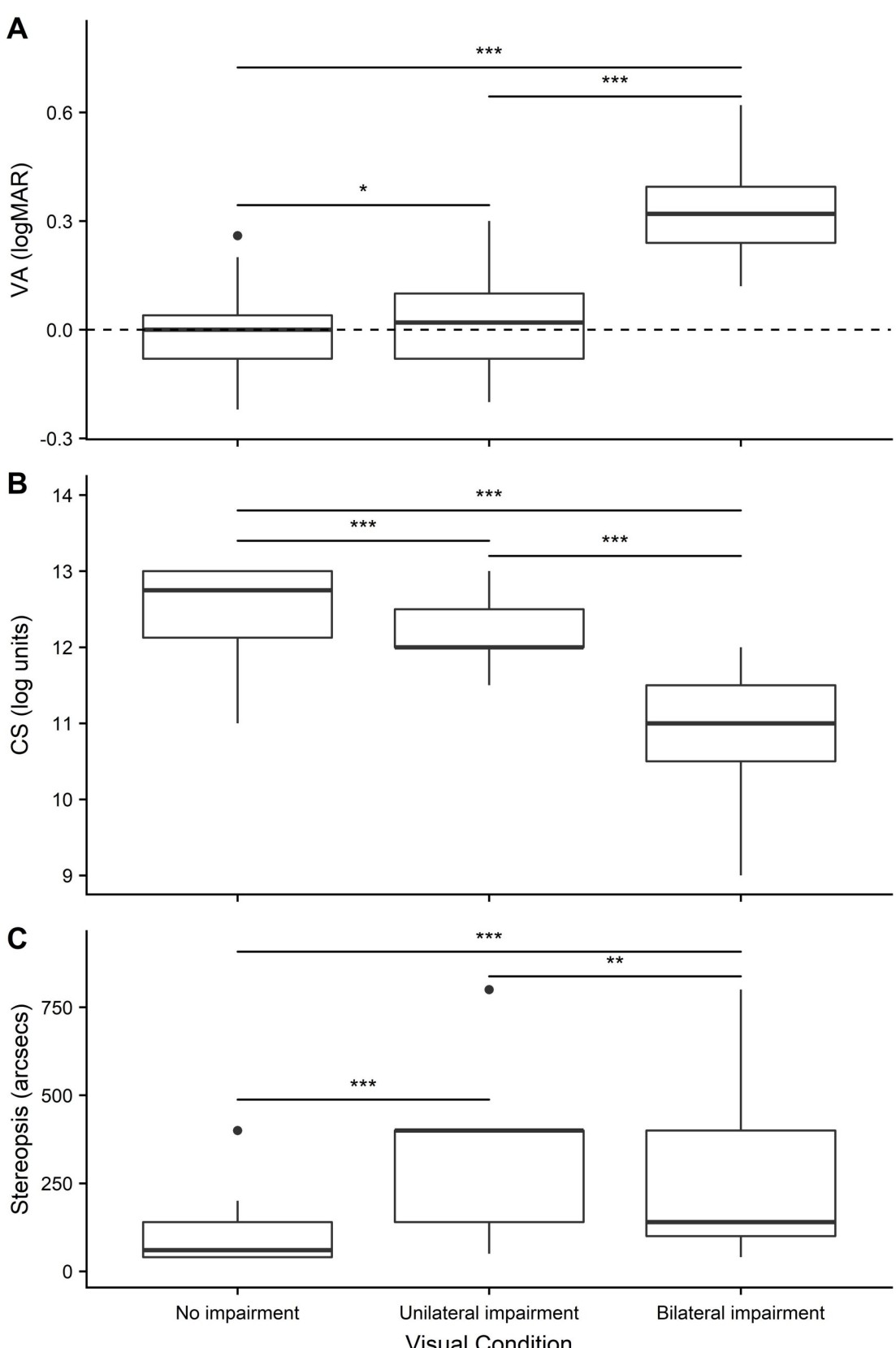

**Fig 3.** A) Visual Acuity (VA; mean logMAR score); B) Contrast sensitivity (CS; mean score in log units); C) Stereoacuity (mean score in arcsecs) across all visual conditions (no impairment, unilateral impairment, bilateral impairment). Significant effects are represented as: $p < 0.05$ (*), $p < .01$ (**), $p < .001$ (***).

A main effect of visual condition was found ($F_{2,58} = 6.70$, $p = .002$, $\eta^2 = .19$). This was driven by a significant difference between no impairment and bilateral impairment ($p = .004$, $d = .65$) and a marginal difference between no impairment and unilateral impairment ($p = .050$, $d = .432$).

*Purdue pegboard.* Pegboard scores (see Fig 5) were best under the no impairment condition and worst in the bilateral impairment condition. A significant main effect of visual condition was found ($F_{2,58} = 4.76$, $p = .012$, $\eta^2 = .14$). This was driven by significant differences in the number of correct placements between no impairment and bilateral impairment ($p = .030$, $d = .502$) and between unilateral impairment and bilateral impairment ($p = .031$, $d = .469$).

*Aiming.* Aiming time (Fig 6) was lowest in the no impairment condition and highest in the bilateral impairment condition. A main effect of visual condition was found ($F_{2,54} = 4.41$, $p = .017$, $\eta^2 = .14$). This was driven by significant differences in movement time between no impairment and bilateral impairment ($p = .032$, $d = .519$).

## Discussion

Experiment 1 assessed the impact of simulated unilateral and bilateral visual impairment on clinical measures of vision and motor task performance. Performance on all visual tests was negatively affected by both unilateral and bilateral filters when compared to normal, unde-graded vision. With bilateral filters impairing VA and CS more than unilateral (as expected), and stereoacuity being most affected in the unilateral impairment condition (again, as expected).

Motor performance was measured using water-pouring, pegboard, and aiming tasks. Performance was reduced on all tasks by bilateral impairment only when compared to baseline (no impairment) or by both bilateral and unilateral impairment. These results are consistent with a large number of previous studies [24,52,53]. Specifically, bilateral impairment (compared to baseline) was found to reduce performance on the pegboard and aiming tasks and for accuracy and a time x accuracy composite measure on the water-pouring task. Unilateral impairment was found to marginally reduce performance in the water-pouring time x accuracy composite. Finally, compared to unilateral impairment, bilateral impairment led to reduced performance on the pegboard task.

It is clear that the ramifications of bilateral impairment are large, but our findings also point towards unilateral impairment being associated with some reduction in motor performance. Previous studies using water-pouring tasks have found no impact of unilaterally degraded vision on accuracy or time measures [24,31]. However, the present study employed a novel technique for analysing these data, presenting a composite accuracy x time measure for water-pouring. The small-medium [54] marginal effect of unilateral impairment on this measure ($p = .050$, $d = .432$) suggests that some aspect of unilateral degraded vision was impacting performance. These findings are most likely due to the overall reduction in vision due to unilateral visual deficit, reduced CS, VA and stereopsis making it more difficult for the participants to see the markings on the jug/beaker and complete the task with speed and precision, suggesting that visual deficits similar to those used in the present study may impair an individual's ability to complete time-bound, precision tasks.

Thus, Experiment 1 established that bilateral and unilateral visual loss produces quantifiable changes in sensorimotor performance in tasks that relate closely to ADLs. Nonetheless, there

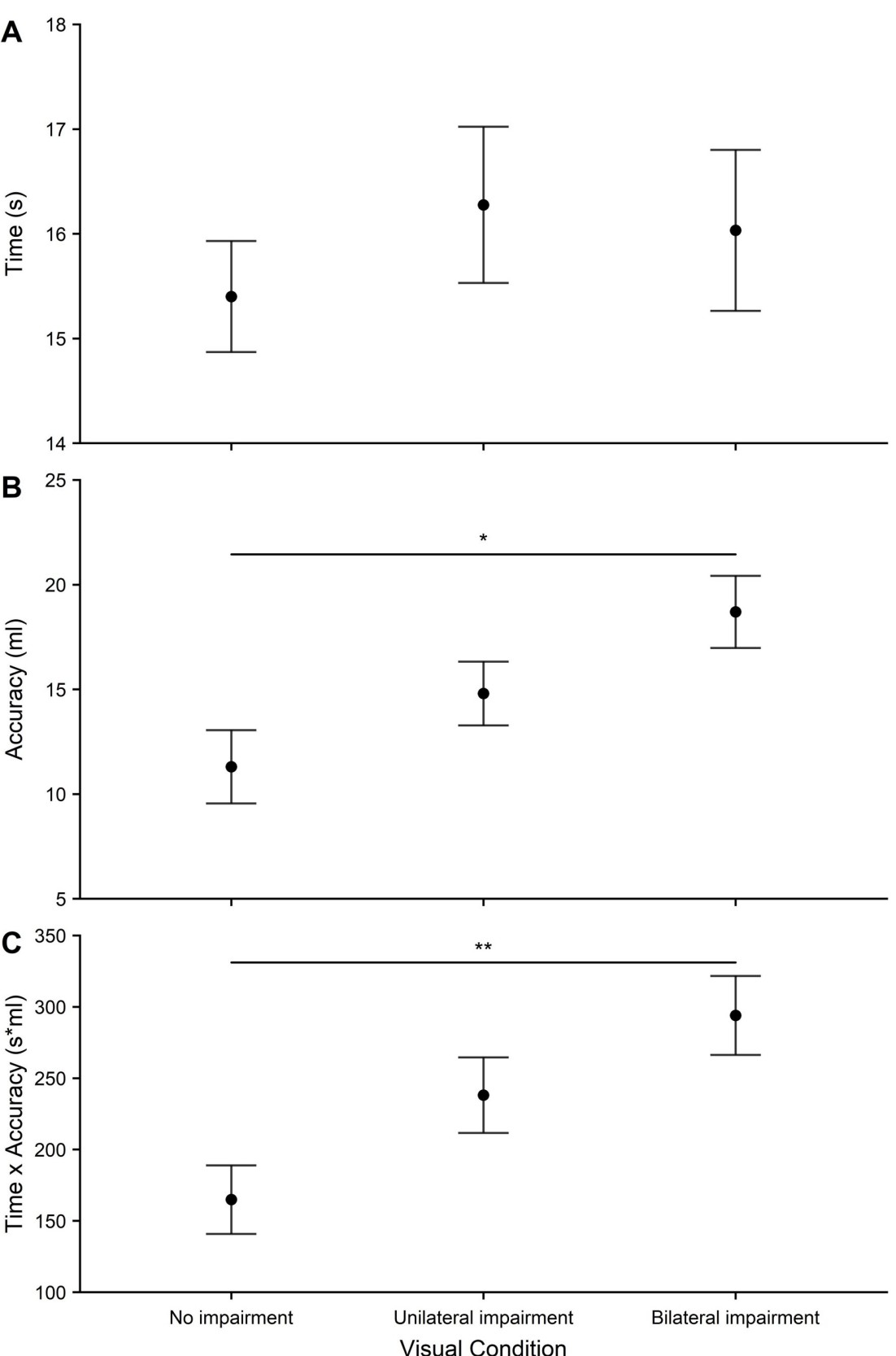

**Fig 4.** Water-pouring measures: A) Time taken (s) to complete the water-pouring task across all visual conditions. B) Water-pouring absolute accuracy (ml). Mean volume of water either above or below the 40ml marker across all visual conditions. C) Water-pouring composite measure of time and accuracy across all visual conditions. Error bars represent the standard error of the mean. Significant effects are represented as: $p < 0.05$ (*), $p < .01$ (**), $p < .001$ (***).

are some limitations to the tasks used here–firstly, they lack standardization (i.e. the weight and shape of the jug may vary between studies, making direct comparisons of results more challenging) and are somewhat coarse measures of performance which may fail to detect the effect of subtle visual impairments such as those used here. Experiment 1 demonstrates that unilateral impairment can reduce motor performance, but also highlights the need for fine-grained kinematic measures of performance (as detailed in the General Discussion).

In addition, in the water pouring and aiming tasks, there were no statistical differences between the unilateral impairment condition and both the no impairment and the bilateral impairment conditions, but the failure to reject the null hypothesis means that these results are difficult to interpret: The lack of a difference between unilateral impairment and no impairment suggests that the unilateral filter has no impact. In contrast, the lack of a difference between unilateral impairment and bilateral impairment suggests that a unilateral filter creates similar impairment to filters in front of both eyes. These results illustrate that exploratory studies have limitations when going beyond the identification of main effects, and highlights the need for hypothesis-driven experiments once exploratory studies have identified a main effect. For this reason, we turned our attention to binocular vs monocular viewing in Experiment 2.

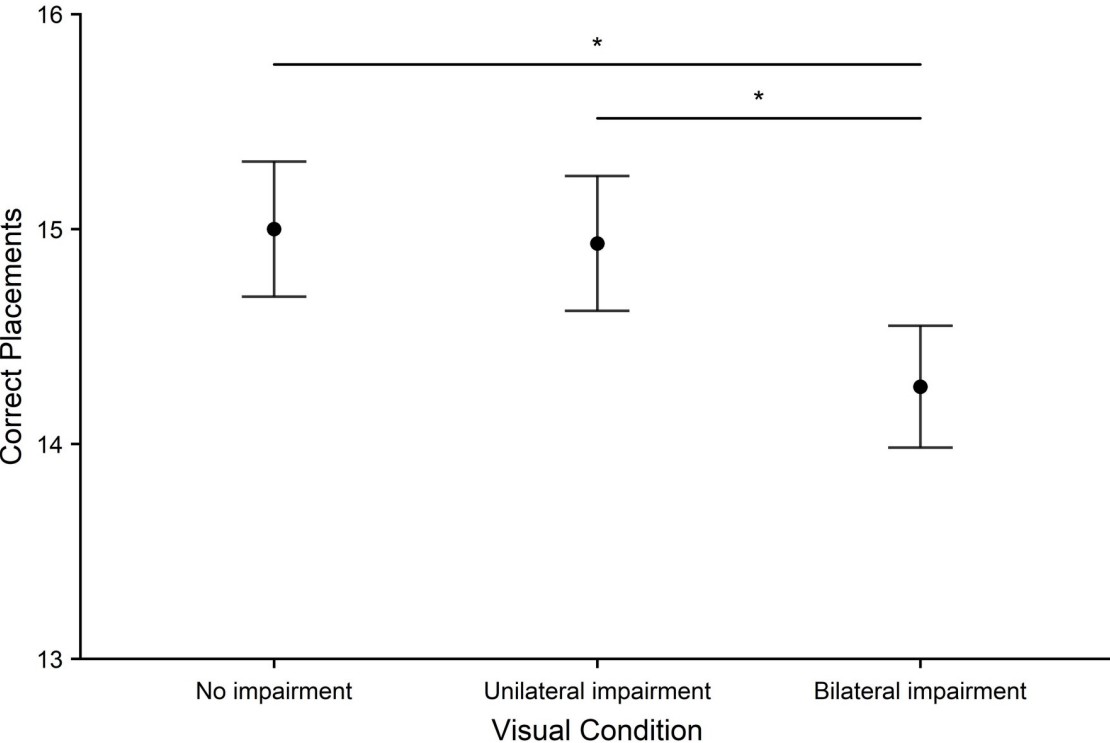

**Fig 5. Purdue pegboard scores.** The mean number of correctly placed pegs within 30 seconds across all visual conditions (no impairment, unilateral impairment, bilateral impairment). Error bars represent the standard error of the mean. Significant effects are represented as: $p < 0.05$ (*), $p < .01$ (**), $p < .001$ (***).

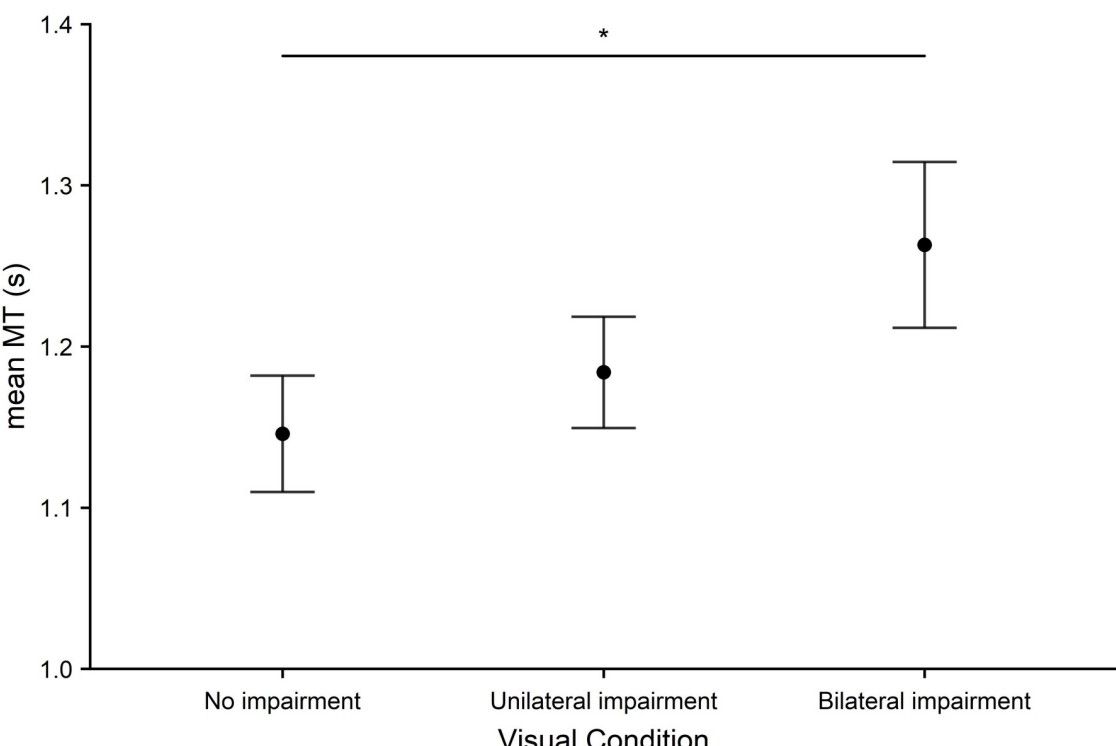

**Fig 6. Aiming.** Mean movement time (MT) in seconds across all visual conditions (no impairment, unilateral impairment, bilateral impairment). Error bars represent the standard error of the mean. Significant effects are represented as: $p < 0.05$ (*), $p < .01$ (**), $p < .001$ (***).

### Experiment 2

In Experiment 2, we sought to use a standardized task to test whether accurate kinematic measures could detect the impact of reduced visual sensitivity, caused by eliminating binocular vision [8–10] and asking participants to perform tasks monocularly, on sensorimotor performance, as seen in tasks such as prehension [20–23]. We hypothesised that monocular viewing would cause a decrease in motor performance on a task where optimal performance requires online visual feedback to implement fast motor corrections.

Many motor tasks (e.g. driving) require visual feedback correction due to the presence of noise (perceptual, neural, and muscular), rendering feedforward control insufficient [35,55]. To test this hypothesis we presented three planar tasks and measured performance under monocular and binocular viewing conditions. Our measures capture critical visuomotor transformations (tracking, steering, and aiming) that underlie many activities of daily living and reflect core factors of real-world skills such as handwriting, postural control and driving. More specifically, the CKAT software used in the present study [46] has been previously been used to investigate fine-motor control across a range of ages including children and older adults [47,56], in healthy and clinical populations [57–59], and has been used to investigate the cognitive underpinnings of skill acquisition [60], as well as the development of specific motor skills such as handwriting and laparoscopic surgery [61,62]. In terms of investigating the impact of degraded monocular vision and motor function, Domkin and colleagues demonstrated that changes to VA predicted changes in performance on a low contrast version of the CKAT aiming task [36]. The portability and scalability of the CKAT battery (see use in large cohort studies such as Born in Bradford [63,64]), make it a simple, objective way to measure changes to sensorimotor function.

The significance of presenting the targets on a plane was that the task did not require the forms of information that arise from binocular viewing per se (e.g. vergence and stereopsis). Likewise, there were no theoretical reasons to suppose that a reduced field of view would negatively affect performance (as the screen was entirely visible even with monocular viewing). The online feedback system needs to detect when the end-point effector (the stylus tip in our experiments) has moved away from the desired path. This means the visual gap between the position of the stylus and the desired path must reach the threshold for detection. In terms of predicting the effects of reduced vision on each of our tasks, the targets presented were all comfortably above the visual threshold, so the parsimonious explanation for any reduced performance would be that, whilst binocular summation activates more neurones, creating a stronger neural signal that might be processed faster [65], monocular viewing creates lags in the online feedback system, delaying the implementation of motor corrections and thereby decreasing performance levels [66].

In other words, the increased sensitivity associated with binocular vision means that the detection of a disparity between actual and desired position will be faster under binocular vision and this will, in turn, allow a more rapid correction of the movement trajectory. Conversely, there will be a relative delay (lag) in response under monocular viewing conditions. Our aiming and steering tasks are particularly vulnerable to this phenomenon as optimal performance relies on fast online corrections [66]. Notably, tracking has less dependency on online visual feedback as optimal performance involves predictive internal models [67] and the coupling of hand and eye movements [68]. It follows that theoretical analysis of the task requirements predicts an effect of viewing condition in the aiming and steering tasks relative to tracking.

## Method

**Participants.**   Seventy-four undergraduate students (55 females) from the University of Leeds participated in this study and were recruited by opportunity sampling. Two participants had to be removed as they did not complete the both eyes condition due to technical failure, leaving a total of 72 participants. Participants were aged 19–24 years old (*mean* = 20.3, *SD* = 1.8). Seven participants were left-handed, 14 had corrected-to-normal vision (using glasses or contact lenses). Participants' VA was measured with both eyes (*mean* = -0.02 logMAR, *SD* = 0.13), with their better eye (*mean* = -0.01 logMAR, *SD* = 0.13), and with their worse eye (*mean* = 0.10 logMAR, *SD* = 0.16). The *mean* difference in VA between the participant's best and worst eye was 0.11 logMAR (*SD* = 0.13). No participants reported any history of tremors or impaired motor function that might affect their ability to perform the tasks. Participants were not compensated for their participation. Informed consent was obtained, and ethical approval was granted by the University of Leeds Research Ethics Committee (Ethics Reference Number: 15–0264) in accordance with the Declaration of Helsinki.

**Materials.**   Fine motor control was measured using the Clinical Kinematic Assessment Tool (CKAT), but this time we used the complete battery rather than just the aiming task. VA was measured using a logMAR chart [39] as in Experiment 1. A VA score was calculated for each of the three visual conditions (two monocular conditions, binocular viewing) from the number of letters correctly identified. VA scores were used to identify the better eye for each participant.

**The sensorimotor battery.**   This battery contained three tasks (tracking, aiming and steering), lasting approximately 12–15 minutes in total. Fig 1 shows a graphical representation of the tasks. Tasks were completed in the following order by all participants.

*Tracking*. Participants were required to keep the stylus within the area of a 10mm green circle moving in a 'figure-of-8' (55mm height, 110mm width) around the screen. The circle

completed nine revolutions of the screen getting faster every three revolutions (producing three levels of speed: slow, medium and fast over 84s). This task consisted of two trials, the first was unguided (No Guide [NG]), and the second was guided (With Guide [WG]). In the WG condition a 'figure-of-8' of 3mm width, the guide was visible on the screen. Mean root-mean-square error (RMSE) was calculated for each speed and guide condition, as the straight-line distance (in mm) from the centre of the moving point to the tip of the stylus. This was recorded at 120Hz and a lower score indicated better performance.

*Aiming.* See Experiment 1 methods for details.

*Steering.* Participants began the task by touching the tip of the stylus in the 'Start' position, they were required to hold this position for 1 second. Participants were then presented with a standard pathway of 4mm width made from two parallel lines. The pathway contained both straights and curves. The participants were required to move the stylus from the 'Start' to the 'Finish' position without removing the stylus from the screen. Speed was controlled using a 'pacing box' overlaid on the track that participants were instructed to stay within. The pacing box covered 1/7th of the track and moved every 5 seconds. There were two different paths (path A and path B). Path B was a vertical mirror image of path A. Participants completed three trials on each, and each trial lasted for around 36 seconds. Path accuracy (PA) was quantised and recorded as the arithmetic mean of the distance between the stylus position and the ideal path (in mm) at each time point through the trial, along with the time taken to complete the path (although this was around 36s if participants remained within the pacing box). To mitigate against the fact that some participants might not have remained inside the pacing box (and completed the task too slowly or too quickly) penalised path accuracy (pPA, mm; as per Flatters et al. [47]) was calculated as:

$$pPA\ (mm) = path\ accuracy\ (mm) * \left(1 + \left(\frac{movement\ time\ (s) - 36}{36}\right)\right)$$

Scores were averaged across the 3 trials of the same path to produce one pPA value for each path.

**Procedure.** Upon arrival, participants were presented with an information sheet, consent was given and demographic, handedness and vision (corrected/uncorrected) information were recorded. Each participant then completed the logMAR test to determine visual acuity in each eye, so the eye with the better visual acuity could be designated the 'better eye'. Next participants completed the CKAT battery under three visual conditions (better eye, worse eye, and both eyes). In the better eye condition, the participant's eye with the worse logMAR score was occluded using an opaque eye patch; in the worse eye condition, the eye with the better logMAR score was occluded. Where participants had equal VA in the right and left eyes, the left eye was assigned as the better eye. Participants completed each task once with no opportunity for practice, however, they were able to ask questions or for tasks to be demonstrated if they were unsure what to do, and the order of vision conditions was counterbalanced to mitigate against practice effects. The order of sensorimotor tests was the same for all participants (tracking, aiming, steering).

**Design and data analysis.** A repeated-measures design was used, with the visual condition (better eye, worse eye, both eyes) as the independent variable for all measures (tracking, aiming and steering). The statistical software package JASP [48] was used to conduct repeated-measures ANOVAs to examine differences between each level of visual condition. For tracking, an additional two independent variables were factored into a 3x3x2 (visual condition x speed x guide) repeated-measures ANOVA. Where Mauchly's test of sphericity was violated, a Greenhouse-Geisser correction was applied. We hypothesised a difference between binocular

and monocular viewing and between the better and worse eye so we tested these predictions using planned pairwise comparisons where main effects were significant, and a Bonferroni Holm correction was applied. The only interaction that emerged in the tracking task did not include visual condition (our primary variable of interest) so we did not investigate further with additional tests due to the focus of this paper.

We first ran analyses with all participants included and these data are displayed in the figures. We were also interested in whether our findings would change when those participants with poorer vision were excluded, so reran the analyses excluding those with a logMAR score over >0.2 in either eye (11 participants) and then again excluding those with an interocular difference >0.2 log units (11 participants) to determine whether patterns remained. For reasons of brevity, we only report the statistics for the second set of analyses when the overall findings differ from the initial analyses. These analyses were performed used RStudio Version 1.3.959 [51]. All data are available from the GitHub database (https://github.com/willsheppard9895/DegradedVisionData.git).

## Results

**Tracking.** Means for tracking are displayed in Fig 7. One participant was removed due to a technical failure, leaving a participant total of 71. No main effect of visual condition emerged ($F_{1.65,115.68} = 1.44$, $p = .24$, $\eta^2 = .02$). There were significant main effects for guide ($F_{1,70} = 5.29$, $p = .024$, $\eta^2 = .07$), and speed ($F_{1.04,72.59} = 235.21$, $p < .001$, $\eta^2 = .77$), and a guide by speed

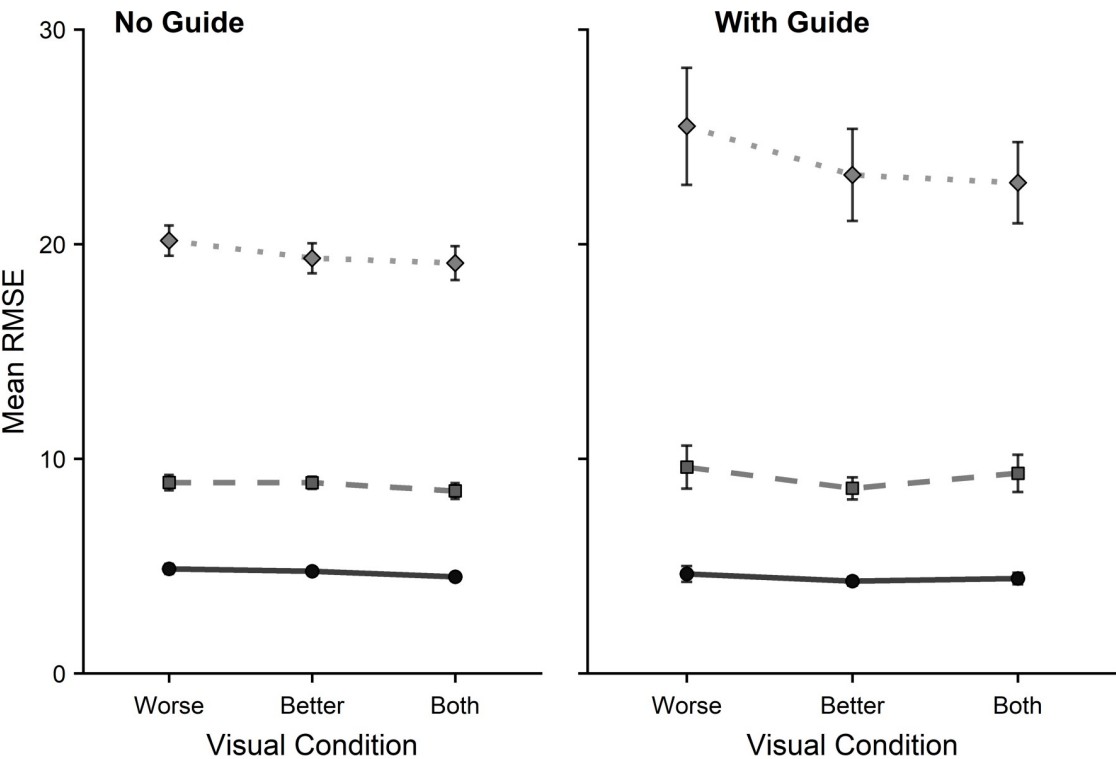

**Fig 7. Tracking a moving target in a 'figure-of-8' trajectory with No Guide (NG) or With Guide (WG).** Performance was measured using the mean root mean square error (RMSE) for all three speeds (Slow = circles, Medium = squares, Fast = diamonds) across all visual conditions (worse eye, better eye, both eyes). Note that the WG conditions were completed second so worse (and more variable) performance may reflect participant mental fatigue/boredom. Error bars represent the standard error of the mean. Where no error bars appear they are smaller than the size of the symbol.

interaction ($F_{1.07,74.65} = 10.10$, $p < .001$, $\eta^2 = .13$) but no interactions involving visual condition. This pattern of results was the same when the participants with a logMAR score over >0.2 in either eye (remaining $N = 60$) or an interocular difference >0.2 logMAR (remaining $N = 60$) were excluded (see *S1 Table* for descriptive statistics for each group).

**Aiming.** Aiming times are shown in Fig 8. Scores were worst in the worse eye condition and best with both eyes. A significant main effect of visual condition emerged ($F_{2,142} = 12.61$, $p < .001$, $\eta^2 = .15$). This was driven by significant differences in time between both eyes and the better eye ($p < .001$, $d = .46$), both eyes and the worse eye ($p < .001$, $d = .56$) but not between the two monocular conditions, ($p = .483$, $d = .08$). As with tracking, this pattern of results was the same when all participants with a logMAR score over >0.2 in either eye (remaining $N = 61$) or an interocular difference >0.2 logMAR (remaining $N = 61$) were excluded (see *S2 Table* supplementary materials for descriptive statistics for each group).

**Steering.** Steering errors (Fig 9) were greatest in the worse eye condition and lowest in the both eyes. A significant main effect of visual condition emerged ($F_{1.38,97.77} = 8.94$, $p < .001$, $\eta^2 = .11$). This was driven by significant differences in scores between both eyes and the better eye, ($p = .011$, $d = .34$), both eyes and the worse eye ($p < .001$, $d = .44$), and between the two monocular conditions ($p = .037$, $d = .25$). A main effect of path was also found ($F_{1,71} = 8.16$, $p = .006$, $\eta^2 = .10$).

This pattern of results was slightly different after excluding participants with a logMAR score >0.2 in either eye. Furthermore, the main effect of path became non-significant ($F_{1,360} = 3.49$, $p = .063$, $\eta^2 = .01$). The main effect of visual condition remained ($F_{2,360} = 5.03$, $p = .007$, $\eta^2 = .03$), but this was now only driven by a significant difference between both eyes and the

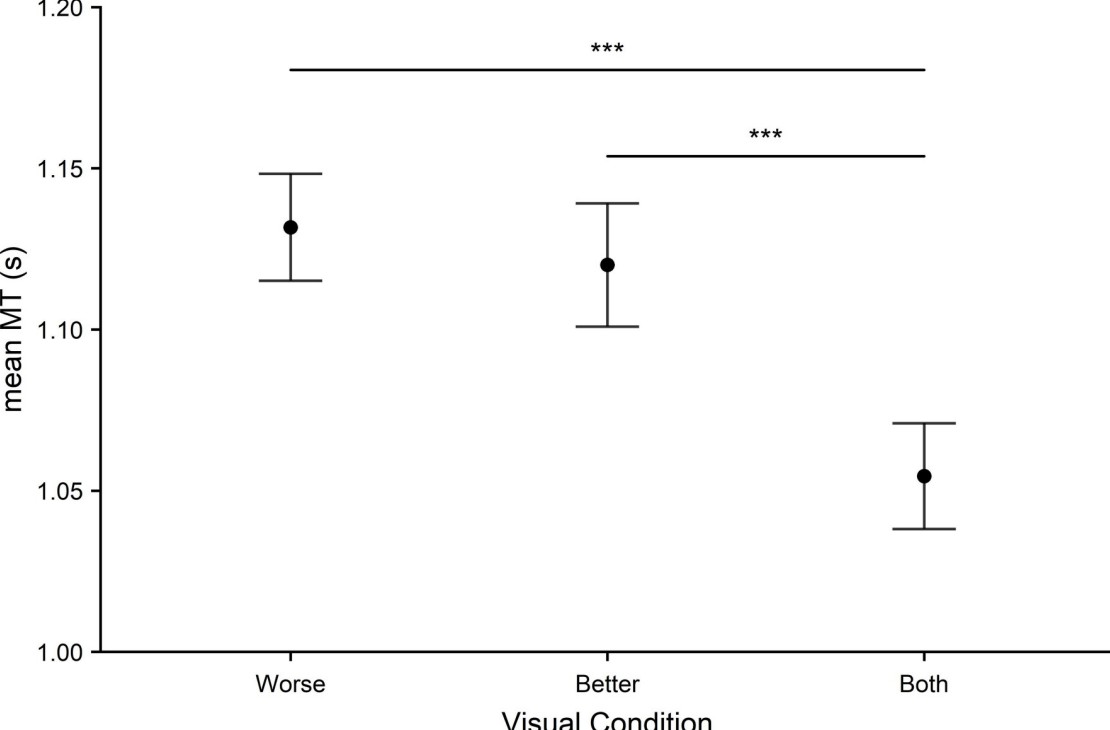

**Fig 8. Aiming performance was measured using mean movement time (MT) in seconds across all visual conditions (worse eye, better eye, both eyes).** Error bars represent the standard error of the mean. Significant effects are represented as: $p < 0.05$ (*), $p < .01$ (**), $p < .001$ (***).

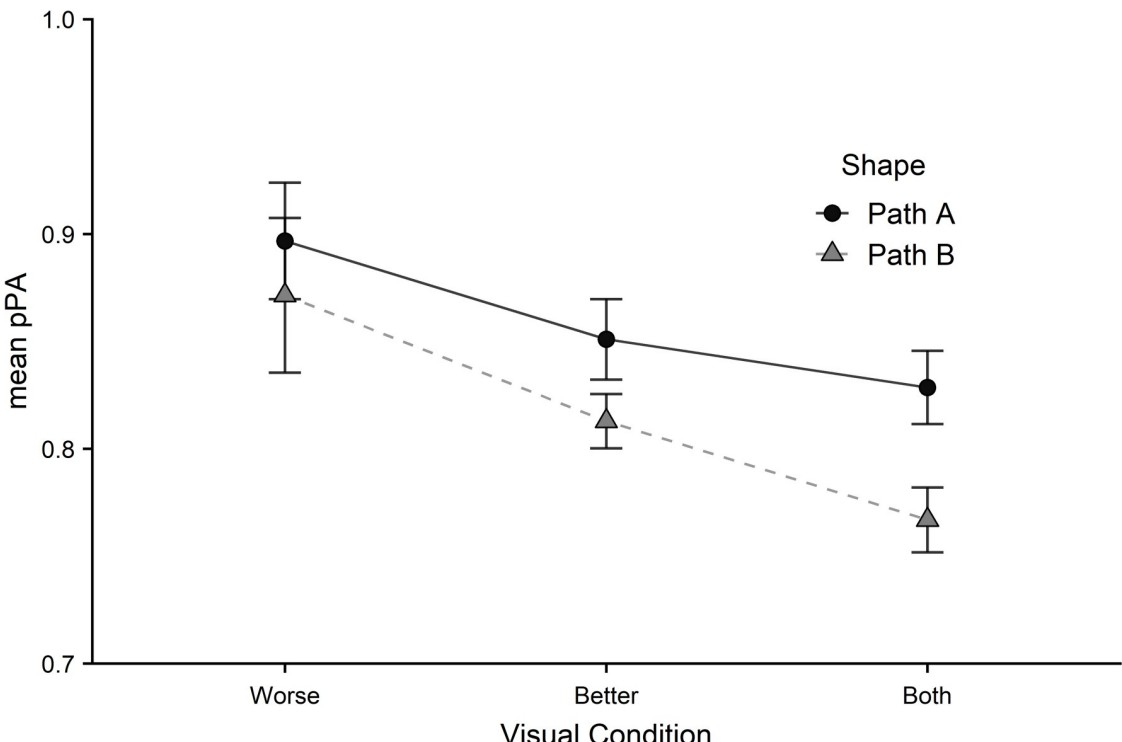

**Fig 9. Steering performance was measured using mean penalised path accuracy (pPA) for both paths (circles = path A, triangles = path B) across all visual conditions (worse eye, better eye, both eyes).** Error bars represent the standard error of the mean.

worse eye ($p = .005$, $d = .36$). There was no difference between both eyes and the better eye ($p = .53$, $d = .19$) or between the two monocular conditions ($p = .103$, $d = .24$).

This pattern of results was also different when excluding participants with an interocular difference of $>0.2$ logMAR (remaining $N = 61$). A significant main effect of visual condition again remained ($F_{1,360} = 5.34$, $p = .005$, $\eta^2 = .03$). This was driven by significant differences in scores between both eyes and the worse eye ($p = .003$, $d = .37$), but not between both eyes and the better eye ($p = .35$, $d = .25$) or the two monocular conditions ($p = .147$, $d = .22$) (see *S3 Table* supplementary materials for descriptive statistics for each group).

## Discussion

In Experiment 2 we sought to determine the impact of eliminating binocular vision on sensori-motor performance. This experiment is the first to investigate whether performance during tracking, aiming and steering tasks (from the standardized 'CKAT battery') is significantly better when using two eyes compared to one and whether performance depends on whether the participant used their better eye or worse eye. In the tracking task, there was no main effect of visual condition, suggesting that monocular vision is sufficient for tracking a moving object. This finding is consistent with *a priori* predictions–suggesting that accurate tracking depends on the prediction of the target movement rather than the online correction of visually signalled errors in skilled adults [67]. In the aiming task, a main effect of visual condition was found (showing an advantage of binocular viewing), but no difference was found between the two monocular conditions. This is most likely due to differences in VA between the two eyes typically being small for most participants (i.e. the visual acuity of the 'worse' eye was not much

worse than the 'better' eye). Evidence for a binocular advantage was also found in the steering task, with an advantage emerging during viewing by the better eye relative to the worse eye. This binocular advantage in the steering task can be considered a robust finding as it remained despite removing participants with poor monocular vision (VA in either eye >0.2 logMAR) or substantial interocular differences (VA difference between eyes >0.2 logMAR) demonstrating the functional significance of binocular viewing even when compared to good monocular viewing. Again, these findings fit with our predictions that aiming and steering would be more vulnerable to the effects of eliminating binocular vision than tracking, due to the greater need to make fast online corrections for optimal performance [66].

## General discussion

This study explored the relationship between unilaterally degraded vision and functional sensorimotor measures. Experiment 1 adopted a typical experimental approach and used unilateral and bilateral Bangerter filters to test performance on a range of visuomotor tasks in the presence of visual loss. Visual function (measured by VA, CS and stereopsis) was reduced when a unilateral filter was applied, with further reductions in VA and CS with bilateral filters. Experiment 1 established that there is a functional impact of visual loss, both bilateral and unilateral, on sensorimotor tasks that relate closely to ADLs, converging with the results of a large number of previous studies [18,20,21,23,24,26,52,69,70]. We argue that the weight of evidence is in the support of the notion that unilateral visual loss has functional significance (i.e. it has an impact on human sensorimotor performance).

The results of Experiment 1 also highlight the limitations of this experimental approach in quantifying the impact of visual loss. First, the results do not indicate the aspect of performance being affected by visual loss. For example, the water pouring task has many components (e.g. grasping the jug, moving the jug to the beaker, monitoring the water flow) and it is not possible to identify whether some or all of these components were impacted by the visual loss. Second, tasks such as 'water pouring' lack the standardization of factors that are known to influence sensorimotor performance. For example, the inertial characteristics of the jug, the texture of the handle and the weight of the beakers will all affect task performance. In the absence of a well-defined sensorimotor task, a robust quantification of the relationship between visual loss and motor performance is not possible. Third, clinical measures (such as the Purdue Pegboard) rely on coarse and categorical measures of performance (e.g. the number of pegs placed in holes). The use of coarse metrics was understandable in the previous millennium but is difficult to justify in 2021 given the available sophisticated kinematic measures that can precisely quantify sensorimotor behaviour.

Experiment 2 tested whether the reduction in vision created by monocular occlusion would impact performance on a battery of tasks that required sensorimotor interactions with targets presented on a planar surface. To the authors' knowledge, this is the first time that this has been done. In line with *a priori* predictions, monocular viewing was associated with impaired performance on two of our three tasks (aiming and steering), but tracking performance was unaffected. These results provide powerful evidence that full binocular vision confers an advantage in tasks that rely on fast, online visual corrections (e.g. driving—where visual errors signify the need for motor responses such as turning the steering wheel [71–73].

Our results, presented alongside a wider body of research [74–76], highlight the potential impact of degraded vision in one eye on sensorimotor performance and the need to precisely quantify the impact of visual loss on sensorimotor performance. Precise quantification of the implications of visual loss will increase the quality of information available to clinicians and patients and allow informed treatment decisions to be made. Our study has made an

important contribution in highlighting the idea that sophisticated kinematic measures can quantify the impact of the degrading visual information, even in young, healthy adults. Having established this, we intend to focus future research on individuals with visual deficits, including those who are stereoblind and unable to benefit from the visual information afforded by normal binocular vision, and in particular older populations, for whom visual deficits are more common. Although it stands to reason that our findings would extend to these groups, our findings do not speak to the issue of whether the reduced sensorimotor performance has a significant impact on ADLs and thereby the quality of life of the participants or whether individuals could adapt and compensate for the loss over a longer period. In addition, we acknowledge that our kinematic measures are somewhat limited (planar movements on a tablet). We selected our measures because they capture critical visuomotor transformations that underlie many activities of daily living and reflect core factors of real-world skills such as handwriting, postural control and driving. Nevertheless, using sophisticated motion capture tools to capture 3D movements (such as reaching to grasp) would be a useful future direction, allowing closer replication of real-world tasks whilst still being objective and finely controlled. We would argue strongly that a programme of work is now required to produce standardized lab-based tasks that can be precisely measured (using kinematic techniques) and then related to ADLs and quality of life in large population and cohort studies.

The significance of our proposed programme of work can be illustrated with regard to cataract surgery within the UK (but we could equally have used amblyopia treatment as an example). In the situation where a patient has bilateral cataracts, the patient is usually offered surgery, with the eye with the poorer vision typically operated on first. The number of patients receiving second eye surgery (SES) in the UK ranges from 21–58% across Clinical Commissioning Groups (CCGs), where surgical restrictions ('managed access') are often based on arbitrary visual thresholds [77,78]. Nevertheless, the lack of a systematic body of work establishing the functional benefits of SES means that NICE guidelines (evidence-based recommendations for health and care in England) rely on the surgeon holding 'discussions with the person about the effect of cataract on their quality of life'. It is difficult to see how a meaningful discussion can be held when there is a dearth of evidence that would allow the surgeon to state with confidence what effects a unilateral cataract may have and the possible benefits of removing the cataract [77,78].

In conclusion, our findings support previous works regarding the functional significance of unilateral visual loss. Our results suggest that even subtly reduced visual sensitivity creates measurable differences in sensorimotor performance. There is now an urgent need to bring together the fields of vision science and motor neuroscience to relate visual status with accurate and precise standardized measures of sensorimotor performance. The development of such a programme of work would enable epidemiological studies to determine–in a scientifically rigorous manner—the relationship between visual loss, sensorimotor performance, activities of daily living and quality of life.

## Supporting information

**S1 Table. Grouped means (sd) [min,max] for tracking data.** Showing participants with normal VA (Normal); participants with high visual acuity (VA > 0.2 logMAR in with eye; High); Those with a large interocular difference (a difference in VA > 0.2 logMAR between their eyes; Different). WG–With Guide, NG–No Guide, RMSE–Root Mean Squared Error. (DOCX)

**S2 Table. Grouped means (sd) [min,max] for aiming data.** Showing participants with normal VA (Normal); participants with high visual acuity (VA > 0.2 logMAR in with eye; High);

Those with a large interocular difference (a difference in VA > 0.2 logMAR between their eyes; Different). MT–Movement Time (s).
(DOCX)

**S3 Table. Grouped means (sd) [min,max] for tracking data.** Showing participants with normal VA (Normal); participants with high visual acuity (VA > 0.2 logMAR in with eye; High); Those with a large interocular difference (a difference in VA > 0.2 logMAR between their eyes; Different). pPA–Penalised Path Accuracy (mm).
(DOCX)

## Acknowledgments

Thanks are also due to Elizabeth Yule, Laura Maxwell, and Sara Peskin for their contribution to recruitment and data collection.

## Author Contributions

**Conceptualization:** Rachel O. Coats.

**Data curation:** Rachel O. Coats.

**Formal analysis:** William E. A. Sheppard, Rachel O. Coats.

**Investigation:** Rachel O. Coats.

**Methodology:** Rachel O. Coats.

**Project administration:** William E. A. Sheppard, Rachel O. Coats.

**Resources:** Rachel O. Coats.

**Supervision:** Rachel O. Coats.

**Visualization:** William E. A. Sheppard, Rachel O. Coats.

**Writing – original draft:** William E. A. Sheppard, Rachel O. Coats.

**Writing – review & editing:** William E. A. Sheppard, Polly Dickerson, Rigmor C. Baraas, Mark Mon-Williams, Brendan T. Barrett, Richard M. Wilkie, Rachel O. Coats.

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
