## [Decision Letter · Decision Letter 0]

14 Jun 2021

PONE-D-21-08524

Exploring the effects of degraded vision on sensorimotor performance

PLOS ONE

Dear Dr. Sheppard,

Thank you for submitting your manuscript to PLOS ONE. After careful consideration, we feel that it has merit but does not fully meet PLOS ONE’s publication criteria as it currently stands. Therefore, we invite you to submit a revised version of the manuscript that addresses the points raised during the review process.

Two expert reviewers have assessed your work. Both find your study interesting and well executed, and provide several constructive points of discussion that should be added to your manuscript. Both reviewers also ask for some clarifications and Reviewer 1 in particular points out several missing methodological details that should be included in the manuscript. Overall, I think it should straightforward to address all reviewer comments, so I look forward to receiving the revised version of this interesting work.

As an additional note, you may find [Maiello et al 2018] relevant to your work and perhaps useful for addressing some reviewer comments (e.g. comment 7 from Reviewer 1). I wish to stress however that I am not asking you to cite this paper I have authored, you should do so only if you find that it improves your manuscript, and this will have no bearing on my final decision.

Maiello, G., Kwon, M., & Bex, P. J. (2018). Three-dimensional binocular eye–hand coordination in normal vision and with simulated visual impairment. *Experimental Brain Research*, *236*(3), 691-709.

We look forward to receiving your revised manuscript.

Kind regards,

Guido Maiello

Academic Editor

PLOS ONE

Journal Requirements:

Additional Editor Comments (if provided):

Reviewers' comments:

Reviewer's Responses to Questions

**Comments to the Author**

1. Is the manuscript technically sound, and do the data support the conclusions?

Reviewer #1: Yes

Reviewer #2: Yes

2. Has the statistical analysis been performed appropriately and rigorously? 

Reviewer #1: Yes

Reviewer #2: Yes

3. Have the authors made all data underlying the findings in their manuscript fully available?

Reviewer #1: Yes

Reviewer #2: Yes

4. Is the manuscript presented in an intelligible fashion and written in standard English?

Reviewer #1: Yes

Reviewer #2: Yes

5. Review Comments to the Author

Reviewer #1: This study examined the effect of impaired vision on the performance of several motor tasks: aiming, pegboard, and water pouring in Exp 1, and aiming, tracking and steering in Exp 2. The goal in experiment 1 was to assess the effects of degraded binocular vision while experiment 2 compared performance during binocular and monocular viewing. This is an interesting study with a large cohort of young participants which provides some insight into how short term disruption in vision affects motor performance. The study is well motivated, the design and analysis are appropriate; however, I have a few comments for the authors to consider to improve their paper.

Experiment 1

1. Provide more details about the method used to select the Bangerter filters. Did all participants wear the same strength of the filter such that VA was ~0.3 logMAR? If there was a range of filters used, provide the details (mean, std dev, range). This could be summarized in a Table. In case the same filter was used, its effect on VA, CS and stereo could also be summarized in a Table (ie report the mean, std dev and range) or present the distributions using a boxplot rather than just the mean and std dev which is currently plotted in Fig 2.

2. In Fig 2 mean stereoacuity in no impairment condition is ~100 arc sec which seems too high in a cohort of visually healthy young individuals. I would expect 40 arc sec based on the clinical test, please explain

3. Was there only one trial per visual condition for the water pouring and pegboard tasks? Were participants allowed to practice the task before data collection?

4. Was performance on the water pouring task measured with a stopwatch to the nearest tenth or hundredth of a second?

5. Please provide a more detailed description of the aiming task: what were the instructions: speed, accuracy or both? what was the distance between the dots? The size of the dots? Duration of the interstimulus interval? Sampling rate? The authors mentioned ‘several kinematic measures’ on Line 238 but only one measure is defined in this section, that is ‘movement time’ – based on the definition provided, it sounds like the measure is ‘response time’ rather than ‘movement time’, please clarify. Was accuracy also quantified?

6. Paragraph on lines 369-376. This paragraph implies that reaching performance is significantly impaired by induced blur, however, deficits in reaching have not been reported in ref 55, that study found differences in secondary saccades. Also, if water pouring performance is affected it’s probably due to difficulty in seeing the marking on the beaker that needs to be filled with water rather than increased uncertainty in the position of the hand (proprioception provides input about the position and movement of the hand)

Experiment 2

7. The authors should provide a more compelling case for selecting a binocular vs monocular viewing rather than the three viewing conditions studied in experiment 1. Given the findings in experiment 1, why were the same conditions not studied for the tracking and steering tasks? The logical link between experiment 1 and 2 is missing. There is an extensive literature that compared motor performance on various tasks during binocular and monocular viewing that has not been mentioned at all. Are the tasks studied here unique and provide additional insight about the contribution of binocular vision to feedforward and feedback control? The authors should explain why they expect that ‘monocular viewing creates lags in the online feedback system’ - line 413.

8. Please provide more details for the tracking stimulus: size of the moving circle, speed, size of the stimulus (ie figure 8 path)

9. Was there a practice trial for any of the tasks?

10. I think that presenting the pPA measure is not sufficient, it is very hard to interpret this score (unless the reliability and validity of using this score has been established in previous published work - please provide a reference). Was the path accuracy quantified using RMSE? Please present the spatial and temporal accuracy scores separately to facilitate interpretation.

11. Why was the left eye assigned as the better eye in cases of equal VA rather than using a test to determine the preferred eye (as was done in exp 1)?

12. In the legend of Fig 7 the authors suggest that ‘fatigue’ contributed to poorer performance on the ‘with guide’ condition at the fastest speed. This task was done first and the whole protocol is <15 min, why would fatigue have such a large effect on this task condition? What is contributing to the fatigue? I doubt it’s muscle fatigue. Perhaps there is an alternative explanation

13. Additional analysis was conducted by excluding participants with reduced VA (n=16) or IOD greater than 0.2 (n=11), rather than excluding the data, consider presenting the means for the different groups in a Table or running an exploratory sub-analysis to compare the performance of the individuals grouped by VA. Or perhaps a regression analysis would provide some insight into the contribution of VA to motor performance during binocular or monocular viewing.

14. Why do aiming and steering tasks require fast online corrections (Line 608)? Is there any evidence in this study supporting this interpretation? Lines 248-250 mention that the last 25 aiming trials have a jumping target, and that these trials were excluded from the analysis. If the authors are interested in fast online corrections, analysis of these trials would actually provide some insight into that question.

15. Perhaps the argument starting on Line 620 could be presented more clearly, when performance declined during monocular viewing it’s due to the poor acuity in that the viewing eye rather than the IOD?

16. Paragraph starting on line 640 acknowledges the limitations of the current study and suggest that kinematics could provide more insight into motor performance which may provide some advantages for clinical assessments. In general, I agree with the authors, but it’s important to also acknowledge that the subset of tasks used in this study is quite limited, only planar movements on a tablet, most of daily activities require interacting with objects in a 3D world. Also, there are many other ‘sophisticated kinematic measures’ – it might be worthwhile to provide a brief discussion of why the selected measures were chosen.

Minor comments

Please proof read and correct grammatical and syntax errors.

Line 227: clarify what ‘incorrectly placed pins’ refers to, were these pins dropped? Or placed in the wrong hole?

Figures should be numbered in the order of appearance, in the current manuscript Fig 6 is referred to first on Line 231. Please provide a more detailed description of the aiming task: what were the instructions: speed, accuracy or both? what was the distance between the dots? The size of the dots? Duration of the interstimulus interval? The authors mentioned ‘several kinematic measures’ on Line 238 but only one measure is defined in this section, that is ‘movement time’ – based on the definition provided, it sounds like the measure is ‘response time’ rather than ‘movement time’, please clarify. Was accuracy also quantified?

Figure 1: correct the order of steps: currently Step 5 comes after Step 1

Reviewer #2: The authors present a well written manuscript examining the effects

of degraded monocular on a series of visual tasks. The demonstrated several

effects comparing monocular vs. binocular viewing conditions and best eye vs. worst eye,

showing that degraded monocular vision usually impairs performance.

The manuscript is well presented, my primary request would be for the authors

to provide motivation in the introduction for studying young people with normal vision

as the mansuscript seems to initially be focused on how this may impact visual impairments

in older individuals might impact quality of life. The results show that even in this healthy

population deficits are often apparent so it stands to reason these may generalise

to an older population with actual visual problems (although as mentioned in the discussion this

may be overcome to some degree with training), further it would be interesting to have same age

comparisons with people who are stereo blind (for another study but worth mentioning in dicussion)

L65 what is the functional impact of "a" unilateral visual deficit?

L144 capitalize name "Purdue Pegboard Task"

L197 delete repeated "consisted of"

L210 do you mean participants who correctly identified just 1 circle and

not "the 1 correct"?

Could you add asterisks highlighting the significant comparisions in the charts

presented in Figures 2-5, 8

L412 what about rivalry? Perhaps performance is made worse due to this.

I am not sure what you mean by creating lag can you be more specific.

For example, rivalry might cause uncertainty about

stimulus location which might slow movements. Is there another mechanism you are thinking of?

Perhaps that binocolar summationgenerates a stronger neural signal that might be

processed faster - can you cite prior research backing this idea?

L431 - Particpants weren't compensated? Were they given course credit please clarify?

6. PLOS authors have the option to publish the peer review history of their article (what does this mean?). If published, this will include your full peer review and any attached files.

Reviewer #1: No

Reviewer #2: No

---

## [Author Response · Author response to Decision Letter 0]

28 Jul 2021

thank you very much for taking the time to review our manuscript, the comments that you left have allowed us to improve the quality the piece and for that i am greatful. Please see the attached response to reviewers file for a more detailed response to your comments.

---

## [Decision Letter · Decision Letter 1]

20 Aug 2021

PONE-D-21-08524R1

Exploring the effects of degraded vision on sensorimotor performance

PLOS ONE

Dear Dr. Sheppard,

Thank you for submitting your manuscript to PLOS ONE. After careful consideration, we feel that it has merit but does not fully meet PLOS ONE’s publication criteria as it currently stands. Therefore, we invite you to submit a revised version of the manuscript that addresses the points raised during the review process.

Both reviewers were satisfied with your revisions. Reviewer 2 has a few remaining comments that should be straightforward to address. Regarding your data availability, I can see and correctly access your github link, so no action is necessary, but you may still wish to highlight your dataset by directly referencing it in your main manuscript text. 

We look forward to receiving your revised manuscript.

Kind regards,

Guido Maiello

Academic Editor

PLOS ONE

Journal Requirements:

Additional Editor Comments (if provided):

Reviewers' comments:

Reviewer's Responses to Questions

**Comments to the Author**

1. If the authors have adequately addressed your comments raised in a previous round of review and you feel that this manuscript is now acceptable for publication, you may indicate that here to bypass the “Comments to the Author” section, enter your conflict of interest statement in the “Confidential to Editor” section, and submit your "Accept" recommendation.

Reviewer #1: All comments have been addressed

Reviewer #2: (No Response)

2. Is the manuscript technically sound, and do the data support the conclusions?

Reviewer #1: Yes

Reviewer #2: Yes

3. Has the statistical analysis been performed appropriately and rigorously? 

Reviewer #1: Yes

Reviewer #2: Yes

4. Have the authors made all data underlying the findings in their manuscript fully available?

Reviewer #1: Yes

Reviewer #2: No

5. Is the manuscript presented in an intelligible fashion and written in standard English?

Reviewer #1: Yes

Reviewer #2: Yes

6. Review Comments to the Author

Reviewer #1: (No Response)

Reviewer #2: Thanks for the revised MS. Overall, I am satisfied with the state of things but have a few small requests.

Also, as far as I could see the manuscript does not include a reference or link to

where the open data can be accessed - I may be in error but please make sure this

is present to meet the PlosOne open data policy

L239 - Please list average estimated visual angle and distance of viewer to the tablet screen.

L235 - PLease clairfy in text and caption that participants only do the aiming task

in Exp 1 and that the other tasks are relevant to Exp 2

L272 - please clarify - I think you mean to say:

All baseline tests for performance with full vision

baseline were recorded first for each participant with no time alloted

for practice as the conditions are easily learned. After baseline recordings,

conditions with simulated visual impairments were counterbalanced.

L335 Is there a need to present the interaction between speed & accuracy?

The findings make sense but I'm not sure what it illustrates that aren't already

evident in Figure 4A & 4B? I'm fine either way but if the authors prefer to keep

can you add a sentence or two highlighting why this is useful and what it adds beyond

time and accuracy alone.

L389 The accuracy x time interaction is mentioned again and it is mentioned as novel

but with no further description of its utility - Please to add some justification

on why it is useful beyond considering time and accuracy alone,

Else please omit as described above.

L473 - Please state the average and std.dev. of the difference between worst and best eye and the sd?

7. PLOS authors have the option to publish the peer review history of their article (what does this mean?). If published, this will include your full peer review and any attached files.

Reviewer #1: No

Reviewer #2: No

---

## [Author Response · Author response to Decision Letter 1]

1 Oct 2021

Dear Editor and Reviewers

We would like to thank you for such useful feedback and comments on our manuscript. We hope we have addressed all concerns below and the revised manuscript. All changes to the manuscript have been made using red font and details of these can be found below, where we respond to each comment in red italics. 

Reviewer 2

Also, as far as I could see the manuscript does not include a reference or link to where the open data can be accessed - I may be in error but please make sure this is present to meet the PlosOne open data policy

Thanks. We have added the link in the paper (see line 303-305, page 16 and line 562-563, page 31).

L239 - Please list average estimated visual angle and distance of viewer to the tablet screen.

We have added this detail in text (see line 242, page 11).

L235 - PLease clairfy in text and caption that participants only do the aiming task

in Exp 1 and that the other tasks are relevant to Exp 2

We have added this detail in the text (see line 249-250, page 11) and in the figure caption (see line 274-276, page 14).

L272 - please clarify - I think you mean to say:

All baseline tests for performance with full vision baseline were recorded first for each participant with no time allotted for practice as the conditions are easily learned. After baseline recordings, conditions with simulated visual impairments were counterbalanced.

Conditions were fully counterbalanced, including the no impairment/full vision condition e.g. not everyone did the full vision/no impairment condition first. Apologies that this was not clear. We have altered the text accordingly (see line 278, 280-281, page 14).

L335 Is there a need to present the interaction between speed & accuracy? The findings make sense but I'm not sure what it illustrates that aren't already evident in Figure 4A & 4B? I'm fine either way but if the authors prefer to keep can you add a sentence or two highlighting why this is useful and what it adds beyond time and accuracy alone.

Thank you. We would prefer to keep it in due to the emerging marginal difference between no impairment and unilateral impairment, so have added a statement about why it’s useful on line 225-227, page 10.

L389 The accuracy x time interaction is mentioned again and it is mentioned as novel but with no further description of its utility - Please to add some justification on why it is useful beyond considering time and accuracy alone, Else please omit as described above.

In addition to the text on line 225-227, page 10 we have also addressed this on line 404-405, page 24.

L473 - Please state the average and std.dev. of the difference between worst and best eye and the sd?

We have added this information on line 481-482, page 28.

---

## [Editor Report · Decision Letter 2]

4 Oct 2021

Exploring the effects of degraded vision on sensorimotor performance

PONE-D-21-08524R2

Dear Dr. Sheppard,

We’re pleased to inform you that your manuscript has been judged scientifically suitable for publication and will be formally accepted for publication once it meets all outstanding technical requirements.

Kind regards,

Guido Maiello

Academic Editor

PLOS ONE
---

## [Editor Report · Acceptance letter]

29 Oct 2021

PONE-D-21-08524R2 

Exploring the effects of degraded vision on sensorimotor performance 

Dear Dr. Sheppard:

I'm pleased to inform you that your manuscript has been deemed suitable for publication in PLOS ONE. Congratulations! Your manuscript is now with our production department. 

Kind regards, 

on behalf of

Dr. Guido Maiello 

Academic Editor

PLOS ONE